# Sorting Nexins in Protein Homeostasis

**DOI:** 10.3390/cells10010017

**Published:** 2020-12-24

**Authors:** Sara E. Hanley, Katrina F. Cooper

**Affiliations:** Department of Molecular Biology, Graduate School of Biomedical Sciences, Rowan University, Stratford, NJ 08084, USA; hanleys2@rowan.edu

**Keywords:** sorting nexins, retromer, endosome, autophagy, ubiquitin, lysosome, proteasome

## Abstract

Protein homeostasis is maintained by removing misfolded, damaged, or excess proteins and damaged organelles from the cell by three major pathways; the ubiquitin-proteasome system, the autophagy-lysosomal pathway, and the endo-lysosomal pathway. The requirement for ubiquitin provides a link between all three pathways. Sorting nexins are a highly conserved and diverse family of membrane-associated proteins that not only traffic proteins throughout the cells but also provide a second common thread between protein homeostasis pathways. In this review, we will discuss the connections between sorting nexins, ubiquitin, and the interconnected roles they play in maintaining protein quality control mechanisms. Underlying their importance, genetic defects in sorting nexins are linked with a variety of human diseases including neurodegenerative, cardiovascular diseases, viral infections, and cancer. This serves to emphasize the critical roles sorting nexins play in many aspects of cellular function.

## 1. Introduction

The integrity of the proteome is essential for maintaining homeostasis as well as coordinating stress response mechanisms. Maintenance of homeostasis is handled by sophisticated quality control systems that ensure that malformed or excess proteins are degraded at the appropriate time and location. Selective proteolysis is largely mediated by the ubiquitin-proteasomal system (UPS) and the autophagy-lysosomal pathway (ALP). In general, the UPS is the primary proteolytic route for misfolded or damaged proteins, and short-lived proteins, having essential functions in many critical cellular pathways, including cell cycle progression and transcriptional regulation [1]. The ALP tackles long-lived proteins, dysfunctional or superfluous organelles, and protein aggregates. As the ALP is upregulated in response to cellular stress (nutrient deprivation, hypoxia, oxidative stress), it is considered a major adaptive mechanism, critical for cell survival following unfavorable environmental onslaughts [2,3].

There are three major types of autophagy: macroautophagy, microautophagy, and chaperon-mediated autophagy (CMA) [4]. Macroautophagy (herein autophagy) is also further classified as being selective or non-selective. In non-selective bulk autophagy, autophagic vesicles randomly engulf portions of the cytoplasm and various cytoplasmic components, primarily in response to starvation signals. Selective mechanisms are predominantly utilized to maintain homeostasis under physiological conditions. Cargos include defective organelles such as mitochondria, endoplasmic reticulum, peroxisomes [5], as well as cytoplasmic protein aggregates [6] and pathogenic intracellular invaders including RNA viruses like SARS-CoV-2 (COVID-19) [7,8,9]. In mammalian cells, ubiquitylation of the cargo is critical for recognition by the autophagic machinery, thereby linking UPS and ALP pathways [10,11].

The third protein quality control system, the endo-lysosomal pathway, is intricately linked to UPS and ALP [12]. Cargos are transported from the plasma membrane to sorting endosomes where their fate is decided. In a ubiquitin-dependent pathway, proteins can be selected for inclusion into intraluminal vesicles (ILVs), that by budding away from sorting endosomes, are ultimately delivered to lysosomes for degradation [13]. Alternatively, proteins can be retrieved from this degradative fate and selected for enrichment in endosomal “retrieval” subdomains, namely the *trans*-Golgi network (TGN), or recycling endosomes. From here they are recycled back to the plasma membrane by the secretory pathway [12,14]. Retrieval of cargos is mediated either by the retromer or retriever complex paired with a sorting nexin, and aided by several complexes (CCC (CCDC22, CCDC93, and COMMD), WASH (Wiskott–Aldrich syndrome protein and SCAR homolog)) and branched actin [15,16,17].

Although initially reported as independent pathways, the UPS and ALP are now known to be interconnected, linked by their common requirement for ubiquitin in substrate targeting. Another common thread is that they all utilize the evolutionarily conserved sorting nexin (SNX) family of proteins to move substrates to different destinations (Figure 1). Underlying their importance, genetic defects in SNXs are linked with a variety of human diseases including neurodegenerative, cardiovascular diseases, and cancer [18,19]. In this review, we will discuss the interplay between sorting nexins and protein homeostasis.

## 2. Classification of SNX Proteins

Sorting nexins are a large group of diverse cellular trafficking proteins, with 10 and 33 members identified in yeast and mammals, respectively [20]. The members are classified into different subfamilies based on the structural arrangements of their scaffolding, enzymatic, and regulatory domains [21] (Figure 2). They all share a common phox homology (PX) domain that binds to phosphoinositide (PI) lipids found in organelle membranes [20,22]. Many SNX family members also contain various other conserved structural domains, BAR and FERM domains being the most prevalent. This modularity confers a wide variety of functions, from signal transduction to membrane deformation and cargo binding. Importantly, sorting nexins are crucial modulators of endosome dynamics as well as autophagic functions.

### 2.1. Lipid Binding PX Domain of SNX Proteins

The canonical 100-130 amino acid phox homology (PX) domain was first identified in the NADPH phagocyte oxidase complex subunits p40phox and p47phox [23]. It is highly conserved and predominantly occurs in sorting nexins. Despite this, the PX domains show little sequence conservation across SNX family members [24]. PX domains all possess the same core fold, consisting of three antiparallel β-strands (β1–β3), followed by three α-helices (α1–α3). Crystal structure analysis has shown that these helices form a loop structure that is required for PtdIns3P binding. Mutations in the loop structure invariably result in the dissociation of PX domain proteins from endosomal compartments [25].

It is well established that different PtdIns decorate different membranes [26]. This has resulted in the concept of a phosphoinositide code, that provides membrane identity within the endocytic system [27]. The PX domains of SNX family members predominantly bind phosphatidylinositol 3-monophosphate (PtdIns3P), a signaling lipid enriched in the early endosome membrane [28,29]. However, SNX proteins can also bind to the other PtdIns phospholipids such as PtdIns(3,4)P2, PtdIns(3,5)P2, PtdIns(4,5)P2 and PtdIns(3,4,5)P3 [24]. 

### 2.2. SNX-PX Proteins

The SNX-PX subfamily consists only of a PX domain. This family includes SNX3 which binds to a multi-protein complex called the retromer that consists of three conserved subunits (VPS26, VPS29, and VPS35) [30,31,32]. Cargo recognition is dependent on *SNX3 binding to the retromer. This* exposes a binding site at the interface between SNX3 and VPS26 for cargos containing the Øx/L/M/V recycling motif (where Ø is a bulky aromatic residue) [33,34]. The PX domain associates with the cytosolic face of early endosomes and recycles its various cargos to the *trans*-Golgi network (TGN) [35]. This is aided by the Ankyrin-repeat protein ANKRD50 [36] and WASH mediated actin polymerization (see Figure 3) [33,34]. Multiple SNX3-retromer cargos have been identified including the Wnt sorting receptor Wntless [30,37], the transferrin receptor [38], and the divalent metal ion transporter Dmt1-II [39]. As such, retromer dysfunction impairs many cellular processes and underlies the pathogenesis of various neurodegenerative disorders. For example, a mutation in VPS35, causes late-onset Parkinson’s disease [40], and microarray studies have implicated the retromer complex in Alzheimer’s disease [18,41].

SNX16 is a unique SNX family protein containing a coiled-coil (CC) domain downstream of the PX domain that deserves a special mention. The PX domain binds specifically to PI3P which is found on early and late endosomes. Recently the recycling of E-Cadherin, which mediates cell-cell adhesion in epithelial tissues, was found to be mediated by SNX16 in a novel mechanism. Here the PX and CC domains form a novel pear-shaped homodimer that interacts with E-Cadherin while simultaneously binding to endosomal membranes [42]. Further studies also discovered that higher order assembly of SNX16 via its CC domain drives membrane tubulation and controls neuronal endosomal maturation [43].

### 2.3. SNX-FERM Proteins (SNX17, SNX27, and SNX31)

The PX-FERM sorting nexins are a sub-group of the PX superfamily. This subfamily has a PX domain and a C-terminal 4.1, ezrin radixin, moesin (FERM) domain with an atypical tertiary structure [44]. They are required for endosomal-to-cell-surface recycling of diverse transmembrane cargos [45]. PX-FERM nexins are further divided into two groups: SNX17 and SNX31. These SNXs are cargo adaptor proteins for a retromer-independent complex called the retriever complex, whereas SNX27 associates with the retromer [46]. The retriever complex localizes to early endosomes and recycles its NPxY/NxxY motif-containing cargo proteins to the cell surface [46]. Through quantitative proteomic analysis, over 120 cell surface proteins, including numerous integrins, signaling receptors, and solute transporters, that require SNX17–retriever to maintain their surface levels have been identified [46]. These include the LDL receptor, amyloid precursor protein, and integrins [44,45,47,48]. SNX17 also recognizes all Human Papillomavirus (HPV) L2 proteins. This interaction aids lysosomal viral escape thereby being crucial for HPV infection [49,50,51].

SNX27 is a unique SNX-FERM protein that contains an N-terminal density 95/discs large/zonula occludens-1 (PDZ) domain. This domain binds PDZ-binding motif (PDZbm)-containing cargo such as the β2-adrenergic receptor [21,52,53]. Quantitative proteomics of the SNX27 interactome has provided an unbiased global view of SNX27-mediated sorting. Here over 100 cell surface proteins, including the glucose transporter GLUT1, the Menkes disease copper transporter ATP7A, various zinc and amino acid transporters, and numerous signaling receptors, require the SNX27–retromer [54]. The FERM domain recognizes Asn-Pro-Xaa-Tyr–sorting signals in transmembrane cargos. Some of these cargo proteins need to be phosphorylated to facilitate binding to SNX27 [44,45,47,48]. SNX27 is highly enriched in the brain. Consequently, cargos include proteins involved in neuronal signaling, such, AMPA receptors [55]. Deficiencies in SNX27 function are associated with Down syndrome [56] and epilepsy [57]. More recently, SNX27-mediated recycling of neuroligin-2 (NL2), a protein required for stabilization of synaptic inhibitory receptors contributes to the regulation of inhibitory synapse composition [22].

### 2.4. SNX-BAR Proteins

The SNX-BAR proteins contain an additional BAR (Bin/Amphiphysin/Rvs) domain that can sense membrane curvature and induce membrane tubulation. The BAR domain is a dimerization motif that forms a rigid cup-shaped structure. This permits the BAR regions to induce membrane deformation, transitioning flat membranes to tubular membrane surfaces [28]. Current models propose that both PX and BAR domains have to be engaged with the membrane to ensure specificity and efficient binding. Mammalian cells possess twelve SNX-BAR family members (SNX1, SNX2, SNX4, SNX5, SNX7, SNX8, SNX9, SNX18, SNX30, SNX32, and SNX33) [58]. Many SNX-BAR proteins (SNX1, SNX2, SNX5, SNX6, and the neuronal SNX32) form heterodimeric complexes. These are critical for endosome-to-TGN retrieval and endosome–to–plasma membrane recycling [59]. Current models suggest that endosome–to–plasma membrane recycling is aided by SNX-BAR association with SNX27 and the retromer, with the PDZ domain of SNX27 as the predominant cargo recognition module [54]. Lastly, SNX-BAR proteins, have retromer-independent roles in autophagic processes, which is discussed in more detail below (Section 6) [60,61].

Until recently the molecular details of how SNX-BAR proteins recognize their cargos have remained elusive. Most of the mechanistic insight was gained from studying the retrograde transport of cation-independent mannose 6-phosphate receptor (CI-MPR) [62]. Interestingly, a wealth of earlier studies, report that CI-MPR recycles to the TGN through direct binding of the CI-MPR tail to the VPS retromer [63,64]. The role of the retromer in CI-MPR recycling has recently been reappraised by two independent studies [65,66]. Surprisingly, both groups concluded that SNX-BAR proteins can directly bind to CI-MPR independently of the core retromer trimer. Furthermore, other proteins that are recycled to the TGN e.g., Insulin-like growth factor 1 receptor (IGF1R) also directly bind to SNX-BAR proteins [65]. Supporting these studies, Simonetti et al. used a SILAC-based proteomic approach to identify a recycling motif, (WLM) [65]. Yong et al extended this work and both identified an additional upstream hydrophobic stretch to the recycling motif and over 70 putative SNX-BAR cargos [58]. Based upon these results, an alternative model has been proposed in which SNX-BARs function as a direct cargo selecting module for a large set of transmembrane proteins transiting the endosome.

### 2.5. Other Domains

Some SNX proteins also contain additional domains including SH3 (Src homology 3), RA (RasGTP effector), and RGS (regulator of G-protein signaling) domains [20,67]. These additional protein-protein binding domains enable SNXs to form homo- or heterodimers and associate with larger protein complexes such as the retromer or autophagy vesicles. More recently, the tetratricopeptide repeat (TPR) protein-protein motif found in SNX21 (a member of the PX-associated B subfamily [21]) was identified as a scaffold for the endosomal recruitment of the Huntington’s disease protein huntingtin (Htt) [68].

## 3. Sorting Nexins and Endocytosis Pathways

### 3.1. The Endocytic Network

Endocytic recycling (outlined in Figure 3) has historically been considered a relatively passive process. Now it is understood to be a highly orchestrated program that plays a major role in cellular homeostasis. How this is conducted at a molecular level is critical as thousands of integral membrane proteins, that regulate many cell functions, journey through this network [69]. Here the SNXs play prominent roles in many aspects of plasma membrane remodeling especially in response to physiological conditions.

Although several distinct endocytic pathways are known, clathrin-mediated endocytosis (CME) is the key process in vesicular trafficking for internalization of transmembrane cargos and their ligands. The process requires the coordinated actions of over 60 different proteins [70]. The arrival and departure of these proteins define the different stages of the pathway that eventually results in the fusion of the internalized vesicles with early/sorting RAB5 GTPase positive endosomes. The pathway has been described in detail in other reviews [71,72].

Early/sorting positive endosomes are formed from primary endocytic vesicles that have undergone homotypic fusion or fused with a pre-existing endosome. These mature into late RAB7 positive endosomes by gradually acidifying the fluid within the endosomal lumen, ending with a pH of 5.5. Both early and late endosomes are characterized by a vacuolar domain that contains ILVs, formed by ESCRT (endosomal sorting complexes required for transport) complexes and enriched in proteins earmarked for lysosomal degradation (see [73] for an excellent review). Early endosomes contain significantly fewer ILV’s than late endosomes and are characterized by a tubular domain that buds from the endosome and ferries their contents to recycling pathways. The formation of late endosomes is dependent upon multiple rounds of cargo sorting and ILV biogenesis, coupled with maturation of early endosomes (also known as the multivesicular endosome or multivesicular body). Upon fusion of the late endosome with lysosomes in a structure named an endo-lysosome, ILV’s and their accompanying cargos are degraded by lysosomal enzymes.

### 3.2. SNX Proteins in Recycling Pathways

The fate of endocytosed proteins is decided upon reaching early endosomes [13]. Cargos destined for lysosomal degradation by ESCRT are earmarked by ubiquitin [73]. Other internalized cargos are recycled by retromer or retriever complexes, aided by the actin-remodeling WASH complex and sorting nexins [74]. There are three distinct forms of retromer, SNX-BAR-retromer, SNX27-retromer, and the SNX3-retromer which have been discussed in detail in Section 2 of this review. By associating with endosomal membranes through interaction with the GTPase RAB7 [75] retromer complexes recycle a wide range of internalized transmembrane cargos from early and maturing endosomes (see Figure 3). The cargos recycle back to the cell surface either by fast or slow pathways mediated by SNX27 and SNX4 respectively. Alternatively, cargo-enriched tubules bud from endosomes and passage through the TGN to the cell surface via the secretory pathway in a process called retrograde transport [76,77]. Importantly, as mentioned above it has now been shown that SNX-BAR proteins mediate retromer-independent retrograde transport of various cargos [65,66]. The retriever complex recycles plasma membrane cargos including integrins and lipoprotein receptors. Here the interaction of the retriever complex with SNX17 is essential for cargo selection. The CCC and WASH complexes aid in the recruitment of the retriever to endosomes [46].

### 3.3. SNX Proteins in Promote Endo-Lysosomal Degradation

The trafficking of cargos by recycling pathways to the plasma membrane is a critical physiological role of sorting nexins. In contrast, some sorting nexins prevent promote lysosomal degradation of their cargo. For example, SNX11 promotes the trafficking of TRPV3 (transient receptor potential vanilloid 3) ion channel from the plasma membrane to lysosomes for degradation via protein-protein interactions [78]. Likewise, SNX1 promotes retromer independent trafficking of protease-activated receptor-1 (PAR1), to the lysosomes [79]. Also, SNX1 and SNX6 facilitate the fate of epidermal growth factor receptor (EGFR) and the tumor suppression p27Kip1 [80,81]. Similarly, SNX4 also regulates the accumulation of BACE1, (ß-site amyloid precursor protein-cleaving enzyme), preventing it from trafficking to the lysosome. This is important as BACE1 is an enzyme involved in proteolytic processing of the amyloid precursor protein, which leads to the formation of the pathological amyloid-β (Aβ) peptide in Alzheimer’s disease [82].

### 3.4. SNX Proteins in Other Pathways

The SNX9 subfamily (SNX9, SN18, and SNX33) are also involved in endocytosis. This subfamily is characterized by containing an SH3 domain at the N terminus, a low complexity domain, and a BAR domain at the C terminus. By regulating dynamin polymerization SNX9 is required for efficient clathrin-mediated endocytosis and SNX18 and SNX9 can compensate for each other [83]. Sorting nexins also interact with cargos outside of endo-lysosomal-TGN pathways (see Figure 1 and Section 6 of this review). Here they play significant roles in autophagy pathways in yeast and mammalian cells [61,84,85,86]. Taken together, this serves to emphasize the diverse roles sorting nexins play in maintaining protein homeostasis.

## 4. Sorting Nexin Cargo Recognition in Yeast

### 4.1. The Yeast Endosome System

In *Saccharomyces cerevisiae*, the makeup of the plasma membrane is adjusted in response to different physiological conditions to maintain homeostasis. This is achieved by the internalization of plasma membrane proteins by endocytosis through clathrin-dependent or independent mechanisms [87]. Some proteins are recycled back to the cell surface either via the TGN or by a recycling pathway originating from the endosome [88]. Others, tagged by ubiquitination, are degraded in the vacuole (the yeast equivalent of lysosomes) [89].

Recently it has been proposed that unlike other eukaryotic species, budding yeast lack early endosomes [90,91]. Instead, cargo-carrying vesicles are initially targeted to the TGN. From here, cargos are either recycled or transferred to late endosomes (also known as multivesicular bodies (MVBs) or pre-vacuolar endosome (PVE) compartments [60]). Late endosomes contain cargo-laden intraluminal vesicles which require ESCRT pathways for their formation. Here ubiquitin plays a key role, as the transport of cargos to the vacuole depends on ubiquitin linkages. Thus, ubiquitination serves both as a signal for endocytosis from the plasma membrane and a specific sorting signal for entry into the vacuolar lumen [13]. In the final step, late endosomes fuse with the highly acidic vacuoles that contain proteases for degradation of the endosomal contents. Recently, very elegant experiments using an engineered fluorescent vacuolar cargo and 4D microscopy have suggested that transfer of material from late endosomes to the vacuole most likely involves “*kiss-and-run*” fusion events [90,92].

### 4.2. The Yeast Retromer Architecture

The yeast retromer (Vps35, Vps26, and Vps29), is an evolutionarily conserved protein coat complex. Significantly, pioneering genetic studies in *S. cerevisiae* initially led to the identification of this trimeric structure, and its cargo Vps10 [32,93]. Vsp10 (vacuolar protein sorting 10), sorts the transmembrane protein receptor carboxypeptidase (CPY) into vesicles at the Golgi [94]. Thereafter CPY-containing vesicles plus Vsp10 are transported to the endosome, which upon maturation, fuses with the vacuole, delivering soluble CPY to the vacuole lumen. Vps10 escapes this fate, being recycled back to the Golgi by the retromer complex, making Vps10 available for additional rounds of CPY sorting. Further studies revealed that the yeast retromer forms a pentameric structure with the Vsp1-Vsp17, SNX-BAR sorting nexin [35]. Within this pentameric complex, the retromer recognizes and mediates the packaging of cargos into endosome-derived transport carriers whereas the sorting nexin mediates endosome recruitment [59,95].

How the retromer coats tubulovesicle carriers remains unclear and is somewhat controversial. This is a critical mechanism to understand as retromer disruption is associated with major neurodegenerative disorders [18]. Studies in yeast by five independent groups using different methodologies show that the retromer interacts with the SNX-BAR dimer through Vps29 and Vps35 independent of Vsp26 [34,96,97,98]. However, Cryo-EM studies of the thermophilic yeast *Chaetomium thermophilum* SNX-BAR protein Vps5, and the retromer report different results. Here Koevtun et al. [99] propose a model in which the SNX-BAR protein, Vps5 interacts only with Vps26. Moreover, they suggest that retromer binding to membranes is dependent on Vps5 from which arches of retromer extend away from the membrane surface. Taken together, retromer assembly in *S. cerevisiae* and *C. thermophilum* may be different.

### 4.3. Yeast Cargo Recognition

It is well understood that retromer complexes selectively recognize their cargos through a recycling sequence. In mammalian cells, this has been defined as (ØX[L/M/V], where Ø is F/Y/W,) [73]. However, in yeast, almost all proteins contain this sequence suggesting a different mechanism for cargo identification. By using mutational analysis of two different retromer cargos, Vsp10 and Ear1, Scott Emr’s group discovered that different sites in the retromer subunit Vps26 are required for their recognition [34]. This suggests a model in which a bipartite recycling signal sequence ensures precise cargo recognition by the retromer complex. These striking results show that the retromer utilizes different binding sites depending on the cargo, allowing this complex to recycle different proteins.

Other sorting nexins also contribute to retromer function [60]. Snx3 is an accessory protein that binds the retromer and recycles cargos from endosomes to the TGN [100]. In yeast, it recognizes relatively few cargos, though a recent systematic genome-wide screen expanded its repertoire [60,86]. One cargo, Neo1, deserves a special mention as its discovery uncovered a previously unknown role for the Snx3-retromer [101,102]. Neo1 is an aminophospholipid flippase, that contributes to the phosphatidylethanolamine asymmetry of endosomal membranes [103]. As anticipated, deletion of the canonical Snx3 recycling motif inhibited the sorting of Neo1 [86]. Moreover, it was also discovered that the sorting of Neo1 by Snx3 is required to recycle other Snx3 cargos. This suggests a model in which Neo1-driven lipid flippase activity promotes vesicle or tubule formation [101]. Similarly, the packaging of human SNX3-retromer cargo, Wntless, also requires Neo1 [102]. Taken together, this suggests that the incorporation of Neo1 into recycling tubules is highly conserved and may influence tubule formation.

Another less well understood sorting nexin that contributes to retromer function is the SNX-BAR protein Mvp1 [60,104]. Mvp1 shares conservation with the mammalian SNX–BAR SNX8, whose function, is involved in endosomal sorting [105,106]. Likewise, *mvp1*∆ cells exhibit defects in retromer-dependent retrograde trafficking [107,108]. Recent structural studies have revealed that the Mvp1 SNX-BAR protein exists as an autoinhibited tetramer in which the PX lipid-binding sites are occluded. The Mvp1 dimer retains membrane-remodeling activity and exhibits enhanced membrane binding. This suggests a model in which unmasking of PX and BAR domains is required for Mvp1 function. As most SNX–BAR proteins are invariably dimeric, this finding adds an intriguing layer of complexity to the regulation of SNX–BAR function.

### 4.4. Retromer-Independent Sorting Nexin Function in Yeast

Snx4, Snx41, Atg20 form two distinct retromer-independent complexes (Snx4-Snx41 and Snx4-Atg20 (Snx42)) that are required for endocytic recycling and selective autophagy. Consistent with these roles they co-localize to the endosome and the pre-autophagosomal structure (PAS) [85,109,110]. Moreover, we and others have shown that after nitrogen starvation, they sequester to the perinucleus where they transport nuclear cargos to the vacuole [84,111] (see Section 6.1).

The most studied cargo of the Snx4-Atg20 complex is Snc1, a plasma membrane-directed v-SNARE, required for the fusion of secretory vesicles with the plasma membrane [112]. It is subsequently retrieved from the plasma membrane by endocytosis and recycled to the Golgi apparatus [85,113]. Two distinct pathways move Snc1 within the cell. Plasma membrane recycling is Snx4 independent and requires F-box protein Rcy1 and the aminophospholipid flippase, Drs2, [114]. Retrograde recycling is Snx4-Atg20-dependent and delivers Snc1 back to the TGN from late endosomes [115]. Taken together, this indicates that multiple pathways can regulate a single SNARE as it cycles through the endo-lysosomal system.

A critical responsibility of Snx4-Atg20 heterodimer is to mediate the endosome-to-Golgi transport of Atg9 [116]. Atg9 is an essential integral membrane protein required for autophagosome biogenesis. Atg27 maintains this Golgi-localized pool of Atg9 [117]. In turn, Atg27 recycling and trafficking are regulated by the retromer and Snx4 [118,119]. More recently, it has been shown that Atg27 is recycled from the vacuole membrane using a 2-step recycling process. First, the Snx4 complex recycles Atg27 from the vacuole to the endosome. Then, the retromer complex mediates endosome-to-Golgi retrograde transport [118]. This is exciting as it represents the first physiological substrate for the vacuole-to-endosome retrograde trafficking pathway.

## 5. Interplay between the Ubiquitin Proteolytic System and SNXs

Ubiquitin (Ub) is a small molecule that covalently attaches to lysine resides on its targets. Ub itself can be conjugated to a second Ub molecule resulting in ubiquitin chains differing in linkage types and lengths [11,120]. This wide variety of Ub modifications can have pleiotropic effects on its substrates [1]. K48-linked ubiquitin chains typically target proteins for degradation by the 26S proteasome [121]. On the other hand, K63-linked ubiquitination typically acts as a signaling event to modify function, such as DNA repair, altering protein-protein interactions, and protein trafficking [122].

During endocytosis, membrane proteins are identified as cargo either as part of a programmed biological response (such as ligand-mediated receptor down-regulation) or as a way to remove aberrantly folded or damaged proteins from the cell surface as a quality control mechanism. Membrane proteins are decorated with Ub on the cell surface and early endosomes to trigger their internalization and endosomal sorting respectively [123,124]. A functional ESCRT pathway is also required. In short, cargos that are tagged with the ubiquitin sorting signal are recognized by ESCRT-0. These are then sequentially handed to ESCRT-I and -II or recruited to the ESCRT-I-II supercomplex before being incorporated into ILVs for delivery to lysosomes [89,123].

### 5.1. Sorting Nexins Regulate UPS Activity

Sorting nexins influence the regulation of proteasome activity and substrate degradation by a variety of different mechanisms. These include blocking ubiquitination of protein substrates, inhibiting ubiquitin specificity factors, regulating protein stability of E3 ligases by either enhancing their recycling or degradation pathways, and degrading inactive or excess proteasome complexes. These are summarized in Table 1. Intriguingly, there are several examples of the relationship of sorting nexins with E3 ligases. In yeast, the E3 ligase specificity factor for Rsp5-dependent ubiquitination, Ear1, is recycled by Snx3 [86]. In mammalian cells, SNX18 is regulated by the E3 ligase Mib1, which indirectly promotes Notch signaling [125]. Likewise, Itch (atrophin-1 interacting protein 4), a member of the NEDD4 family of E3 ubiquitin ligases, ubiquitylates SNX9, thereby regulating intracellular SNX9 levels [126]. In a seminal discovery, the E3 ubiquitin ligase partner of MAGE-L2, a protein that enhances E3 ubiquitin activity [127], was found to be K48 E3 TRIM27 [127]. The MAGE-L2-TRIM27 complex localizes to endosomes through interactions with the retromer. The outcome of this interaction is K63 ubiquitination of the WASH complex, a known regulator of retromer-mediated transport. This action permits WASH to nucleate endosomal F-actin (see Figure 3). Moreover, this pathway is regulated by the deubiquitinating enzyme USP7 [128].

Less is known about sorting nexins and E2 activity. In *Drosophila*, UBC-13, the E2 ubiquitin-conjugating enzyme that generates K63-linked ubiquitin chains, is essential for retrograde transport of multiple retromer-dependent cargos, including MIG-14/Wntless. Here UBC-13 function is required for retrograde transport of SNX1 retromer-dependent cargos [135].

In the budding yeast, nitrogen starvation triggers the disassembly of nuclear 26S proteasomes into 19S and 20S subcomplexes. The subcomplexes are consequently transported through the nuclear pore complex (NPC) and targeted to autophagosomes for degradation [84,136,137]. Intriguingly, nitrogen starvation triggers cytosolic Snx4-Atg20 and Snx4-Snx41 heterodimers to relocate to the perinucleus. Moreover, both these heterodimers and the core autophagic machinery are required for 19S and 20S vacuolar proteolysis [84]. This has led to the model that these heterodimers mediate the transport of the proteasome subcomplexes to the phagosomes. How Snx4-Atg20 and Snx4-Snx41 interact with the proteasome subunits remains unknown, but it adds another example of how sorting nexins regulate the UPS machinery.

### 5.2. Sorting Nexins Can Be Regulated by the UPS

In addition to regulating UPS activity, sorting nexins themselves are regulated by the UPS. Although there are significantly fewer examples of this activity reported it still serves to demonstrate the reciprocal nature of the relationship between sorting nexins and the UPS. In short, sorting nexins have been reported to be stabilized by interacting deubiquitinating enzymes (DUBs) [138]. The hormone vasopressin increases the expression of the DUB USP10 that deubiquitinates and stabilizes SNX3 [129]. SNX27 interacts with OTULIN (OTU Deubiquitinase With Linear Linkage Specificity) that specifically hydrolyzes methionine1 (Met1)-linked ubiquitin chains. This antagonizes SNX27-dependent cargo loading and binding of SNX27 to VPS26A and affects endosome-to-plasma membrane trafficking. Moreover, these findings define a non-catalytic function of deubiquitinases in sorting nexin function [132].

## 6. Sorting Nexins in the Autophagy-Lysosomal Pathway (ALP)

Autophagy is considered the first line of defense in response to many forms of extracellular stress [11,139]. Sorting nexins engage in non-selective autophagy pathways following unfavorable physiological cues in both yeast and mammalian cells. In yeast, SNX-BAR heterodimers also are required for efficient selective autophagy pathways.

### 6.1. Sorting Nexins in Non-Selective Autophagy in Yeast

In physiological conditions, the primary function of sorting nexins within the endosomal pathway is to maintain steady-state levels of membrane proteins. Following different stress cues such as nutrient depletion or starvation, SNXs engage in stress-induced regulatory roles that require sorting nexin cellular relocalization. The role of sorting nexins in autophagy was first identified in yeast where it was found to be a component of the Atg1 initiation complex [109]. Here Snx4 binds the Atg17 scaffold complex which is required to localize the PAS to vacuole outer membranes [140,141]. Consistent with this, deletion of Snx4 results in insufficient PAS formation coupled with a delayed autophagic response [142]. In a more recent phosphoproteomics study, Snx4 was identified as a direct substrate for Atg1 [143]. This phosphorylation event may direct Snx4 away from its physiological functions in endosomal sorting and towards its role in starvation-induced autophagy. Interestingly, during selective autophagy, Atg20 interacts with the scaffold protein Atg11 that replaces Atg17 and initiates autophagosome assembly at the cargo site [140]. SNXs also play a role in autophagosome and vacuolar membrane fusion. Here the Snx4-Atg20 heterodimer promotes non-selective autophagy by exporting lipids from the vacuole which maintains its fusion competence [144].

### 6.2. Sorting Nexins in Non-Selective Autophagy in Mammalian Systems

The biogenesis of the phagophore is not fully understood. It is believed to originate from the ER and expand by receiving membranes from different sources including the plasma membrane and Golgi [145]. Two essential proteins needed to build phagophores, ATG16L1 and ATG9A, traffic from the plasma membrane to recycling endosomes on their way to sites of autophagosome formation [146,147,148]. New insight into how this is achieved has come from Anne Simonsen’s group. They revealed that the PX-BAR-containing protein SNX18 recruits Dynamin-2 to induce budding of ATG9A and ATG16L1 containing membranes from recycling endosomes to sites of autophagosome formation (Figure 4A) [149,150]. More recently, the SNX4-SNX7 heterodimer has been shown to play a role in phagophore biogenesis and ATG9A distribution, although the precise molecular mechanisms remain unknown [61]. Given the dynamic nature between organelles during autophagosome biogenesis, it is likely that future studies may reveal similar roles for other sorting nexins.

### 6.3. Sorting Nexins in Selective Autophagy in Yeast

In yeast, Snx4 and Atg20 are required for several selective autophagy pathways. Cargos include mitochondria [151], peroxisomes [152,153], proteasomes [84], ribosomes [84], fatty acid synthase complexes [154] and transcription factors [111] (Figure 4B). A role for sorting nexins in selective autophagy was first identified while studying the cytoplasm-to-vacuole targeting pathway (CVT).

This biosynthetic pathway functions in physiological conditions to transport the aminopeptidase, Ape1, to the vacuole. The Snx4-Atg20 heterodimer is required for the recruitment of proteins to the site of CVT formation [109]. The molecular details of Snx4′s role in this pathway remain unclear. Lastly, a role for sorting nexins in mammalian selective autophagy pathways has not yet been reported.

### 6.4. SNX5 and Viral Autophagy Induction

It is well established that inactivation of the target of rapamycin complex 1 (TORC1) induces autophagy [155]. In pioneering studies, Xiaonan Dong and colleagues have demonstrated that viral-induced autophagy is distinct from previously described selective autophagy and basal autophagy activated by nutrient deprivation and mTOR suppression. Following viral infection, SNX5 initiates autophagosome biogenesis by localizing to virion-containing early endosomes. Mechanistically SNX5 interacts with beclin1 and ATG14- containing Class III phosphatidylinositol 3-kinase (PI3KC3) complex 1 (PI3KC3-C1). SNX5 is also required for increasing the kinase activity of PI3KC3-C1, generation of endosomal PI3P, and recruitment of WIPI2 to virion-containing endosomes. Deletion of SNX5 in mice models results in increased susceptibility to multiple human viral infections and enhances lethality after infection. These exciting results reveal that SNX5 thereby plays an important role in the immune response following viral infections [156].

## 7. The Interplay between Sorting Nexins, Lysosomal Degradation, and UPS-Mediated Degradation

In recent years, it has become apparent that the UPS and autophagy pathways are functionally interconnected [157,158,159]. Key findings from these studies have revealed that when the UPS is overwhelmed, autophagy is upregulated to eliminate aberrant proteins [11]. Furthermore, ubiquitination is utilized as a degradation signal by autophagy pathways, being critical for removing damaged mitochondria by mitophagy in mammalian cells [160,161]. Ubiquitin is also required for autophagic degradation of protein aggregates [162,163], peroxisomes [164] pathogens [165] and ribosomes [166,167]. Here the crosstalk between ubiquitination and autophagy is provided by autophagic adaptor proteins (or autophagy receptors), which bind both ubiquitin and autophagy-specific ubiquitin-like modifiers of Atg8 and its homologs [168,169]. This has led to the more current hypothesis that UPS and autophagy pathways constitute a single integrated degradation system [140]. Consistent with this, following TORC1 inhibition, in yeast, nuclear proteasomes are disassembled and then destroyed by Snx4-Atg20 and Snx4-Atg41 mediated autophagy [84,137,170].

### 7.1. Environmental Cues Dictate the Degradative Fate of Med13

There are a limited group of proteins that are both ALP and UPS substrates. Our group has discovered that in *S. cerevisiae,* Med13, a member of the conserved Cdk8 kinase module (CKM) of the mediator complex, is degraded either by a novel Snx4-mediated autophagy pathway or by the UPS in response to cell survival and death signals respectively (Figure 5) [111,171,172]. The CKM interacts with the mediator complex of RNA pol. II to predominantly repress genes induced by environmental stress [173,174,175]. Activation of these genes is achieved by disrupting the CKM association with the mediator [174,176]. We have shown that Med13 and cyclin C are both targets of the UPS system following oxidative stress [171,172,174,177]. Importantly, before cyclin C is destroyed it translocates to the mitochondria where it mediates stress-induced mitochondrial fission and promotes cell death in both yeast and mammalian cells [178,179,180]. In mammalian cells, mitochondrial located cyclin C also associates with Bax to promote its activation [179].

In contrast, following a survival cue (nitrogen starvation), cyclin C is rapidly destroyed by the UPS before its nuclear release. This promotes cell survival by preventing mitochondrial fission and upregulating AuTophaGy (*ATG*) genes [181]. To our surprise, we discovered that instead of being targeted by the UPS, here Med13 is destroyed by vacuolar proteolysis. After transitioning through the nuclear pore complex (NPC), Med13 is transported by the Snx4-Atg20 heterodimer to Atg17-initiated phagophores at the vacuole [111]. Moreover, two transcriptional activators (Rim15 and Msn2) that positively regulate *ATG* expression, are degraded upon nitrogen starvation by this mechanism. Taken together, this suggests a model in which Snx4-mediated autophagy of *ATG* transcriptional regulators allows fine-tuning of the autophagic response. Moreover, it outlines a new autophagy pathway by which transcription factors are selectively targeted for degradation.

### 7.2. p27 Is Regulated by Proteasome Degradation and SNX6-Mediated Endo-Lysosomal Pathways

It is well established that proteasomal degradation of the growth suppressor p27 regulates cell cycle progression. Proteolytic degradation occurs via two different pathways. In G1 this cyclin-CDK (cyclin-dependent kinase) inhibitor is degraded in the cytoplasm by the E3 ligase Kip1 whereas its degradation at the G1/S transition and in G2 occurs in the nucleus, mediated by the E3 ligase SCF^Skp2^ [182,183]. Cytoplasmic p27 can also be recognized by SNX6 for endo-lysosomal degradation. This is also important for cell cycle progression as silencing SNX6 delays S-phase entry in starvation-synchronized NIH-3T3 cells [81]. It remains unknown if the SNX-mediated endo-lysosomal pathway interacts with p27 proteasomal degradation but it is important to determine as this cyclin-CDK inhibitor not only inhibits the catalytic activity of cyclin D-, E-, A-, and B-CDK complexes but also regulates other processes including cell migration and development independent of its CDK inhibitory action [184].

## 8. Sorting Nexins in Disease

The SNX family consists of a diverse group of proteins involved in various aspects of protein trafficking. Underpinning their importance in protein homeostasis, the etiology of many diseases such as cancer, cardiovascular and neurodegenerative diseases is linked to the dysregulation of sorting nexin function [185]. Moreover, viruses and bacteria can hijack sorting nexin pathways to both invade and replicate in host cells.

### 8.1. The Role of Sorting Nexins in Cardiovascular Disease

Sorting nexins are implemented in the development of cardiovascular diseases such as hypertension, coronary heart disease, and heart failure [19]. Here SNXs influence the maintenance of blood pressure by regulating the expression and function of G-protein coupled receptors (GPCRs) such as dopamine receptors, ion channels, and transporters [19,129,186]. Consistent with this, knockdown/knockout animal models of SNX1, SNX5, and SNX19 correlate with hypertension [187,188]. This has led to addressing if SNXs could potentially be therapeutic targets for hypertension. Therapeutic strategies have focused on expressing specific SNX subtypes within the kidney to decrease blood pressure [19].

SNXs also influence the pathogenesis of coronary artery disease by regulating lipid metabolism. SNXs interact with the leptin receptor and the low-density lipoprotein (LDL) receptor [22,189] and decreasing SNX1 levels results in increased levels of triglycerides and cholesterol [22,189]. SNXs may also influence coronary artery disease by regulating inflammation, which is linked to the etiology of vascular diseases [190]. SNX13 deficiencies correlate with decreased heart function associated with cardiomyocyte apoptosis. SNX13 mediates the recycling of the apoptotic repressor, ARC. Thereby, loss of SNX3 results in the degradation of ARC and promotes cardiomyocyte apoptosis and heart failure [191].

Insulin insensitivity is a major hallmark of type 2 diabetes mellitus which is a characteristic feature of heart failure [192]. Sorting nexins are linked to this pathophysiology as SNX5, SNX19 and SNX27 regulate insulin degradation, secretion, and signaling. Silencing SNX5 in animal models increases blood insulin, decreases insulin excretion, and causes insulin resistance [54,193,194].

### 8.2. The Role of Sorting Nexins in Neurogenerative Diseases

SNX dysregulation has been linked to several neurodegenerative diseases such as Alzheimer’s disease (AD), Parkinson’s disease, and Down syndrome [195]. In neuronal cells, the composition of the cellular membrane is essential for responding to extracellular stimuli and neuroplasticity. SNX-mediated regulation of the cellular membrane composition influences several processes such as neuronal excitability, plasticity, neural development, signaling, psychostimulant response, and cellular drug resistance [196].

Best described is the role of sorting nexins in the pathogenesis of Alzheimer’s disease (for details see [197]). AD is characterized by brain accumulation of extracellular neuritic plaques containing deposits of β-amyloid peptide (Aβ) and neurofibrillary tangles compromised of the microtubule-associated protein tau. One of the proteins involved in regulating the β-amyloid peptide is the SNX33 retromer. This complex inhibits endocytosis of amyloid precursor protein (APP) which in turn leads to retention of APP at the plasma membrane which promotes plaque formation [198]. SNX15 and SNX17 also regulate APP processing [199]. SNX4, SNX6, and SNX12 have been shown to regulate BACE1 trafficking which also controls Aβ peptide generation [200,201]. In addition, SNX27 binds and inhibits γ-secretases thereby decreasing Aβ peptide [202].

Parkinson’s disease (PD) is defined by the loss of dopaminergic neurons and the accumulation of α-synuclein-enriched Lewy bodies. Genome-wide association studies have identified various mutations that increase Parkinson’s disease susceptibility such as PINK1 and Parkin whose gene products regulate mitophagy [203,204]. As such mitochondrial defects such as disruptions in mitochondrial fission and mitophagy are hallmarks of PD. It can be speculated that dysregulation of SNXs may perturb autophagy pathways that are necessary to clear α-synuclein aggregates and damaged mitochondria. In support of this, the pathophysiology of Parkinson’s disease is linked with a mutation in the retromer subunit VPS35 [195]. This is relevant as VPS35 mutations result in decreased association with the WASH complex which perturbs ATG9 transport, ultimately compromising autophagosome biogenesis [205,206]. Lastly, VPS35 also interacts with the mitochondrial fusion regulator, Drp1 [207]. VPS35 mutations are linked to increased mitochondrial fragmentation and cell death [208]. In neurons, this is particularly devastating as mitochondrial fission directs mitochondrial transport to their potential docking sites in axons and dendrites [209]. Lastly, the etiology of PD has also been linked with Pink1 and Parkin function and SNX9 mediated degradation of mitochondria in a vacuolar pathway distinct from mitophagy [210].

SNX deficiencies have been implemented in Down syndrome [56] as well as associated with epilepsy, developmental delays, and subcortical brain abnormalities [196]. SNX27 knockdown/knockout animal models or human patients with non-functional SNX27 variants exhibit a wide range of neurological aberrations that may be associated with defects in cell surface receptors [57]. Some of these receptors include neuroreceptors (AMPA, NMDA), ATPase copper transporters, glucose transporters, disintegrin, metalloproteinase, and adhesion proteins (NLGN2) [196]. For example, SNX27 expression is downregulated in human Down’s syndrome brains. Mechanistically, SNX27 may regulate the retention of cell surface membrane proteins such as the myelination-related protein, GRP17 which plays an important role in oligodendrocyte development [211].

### 8.3. Oncogenic Roles of Sorting Nexins and the UPS

In recent years oncogenic roles of sorting nexins have been reported. Therefore, it comes as no surprise that many of these roles lead to the activation of well-characterized oncoproteins. Recently, TRIM27 has been classified as an oncoprotein. Consistent with this role, it is overexpressed in many cancers, including breast, endometrial, ovarian, lung, and colon [212]. TRIM27 associates with the retromer and activates the cytoplasmic transcription factor, STAT3 [133]. This is an important discovery as STAT3 plays a central role in various physiological processes and its aberrant and persistent activation results in serious diseases, including cancer [213]. It is a cytoplasmic transcription factor as its activation and translocation to the nucleus is dependent upon its passage through the endosome system [214]. In response to several cytokines or growth factors including interleukin-6 (IL-6), STAT 3 is phosphorylated [215]. This promotes its release from the endosome and translocation to the nucleus, resulting in the induction of downstream effector genes. Intriguingly, the E3 ubiquitin ligase activity of TRIM27 is dispensable for its ability to mediate STAT3 activation. Confirming a retromer linked role, knockdown of each of the retromer components significantly inhibits IL-6-induced transcription of STAT-dependent genes [133]. It is well established that endocytosis is an effective mechanism to downregulate cellular signaling events by internalizing receptors or ligand-receptor complexes [216]. Further studies are needed to address if other signaling proteins that are imprisoned by endocytosis, e.g., the promiscuous kinase glycogen synthase 3 beta (GSK3-β), are similarly regulated. This is important as this kinase has numerous phosphorylation targets in distinct pathways, including WNT, Hedgehog, and MAPK signaling [217].

Sorting nexins interaction with E3 ligases plays a role in oncogenesis in other cancers. In head and neck squamous cell carcinoma (HNSCC) SNX5 interacts with the E3 ligase F box proteins, thereby blocking FBW7 mediated ubiquitination of oncoproteins including c-Myc, NOTCH, and cyclin E1 [130]. SNX16 also has oncogenic properties in colorectal cancer, where it is significantly upregulated. This affects eEF1A2/c-Myc signaling, possibly by inhibiting proteasome-dependent ubiquitination of eukaryotic translation elongation factor 1 A2 (eEF1A2) [131]. As such, SNX16 has been implemented in the development of other tumors such as bladder and ovarian cancer [218,219]. SNX10 also may be a tumor suppressor in mouse models of colorectal cancer. Here SNX10 deficiency prevents the degradation of LAMP-2A, the essential CMA lysosomal receptor [220,221]. Given the key role of soring nexins in many biological processes, there is no doubt that future work will reveal more links to cancer and other diseases.

### 8.4. Viruses Can Hijack Sorting Nexin Pathways

In the last few years, it has become apparent that viruses can exploit retromer-mediated trafficking for their replication. Although many details remain unclear it has emerged that the strategies used are diverse. For example, some viral effectors recruit retromer components to viral replication sites to promote infection. An excellent example is the NS5A protein from the hepatitis C virus which interacts with VPS35 [222]. Others may mimic retromer cargo to travel i.e., hitchhike, from endosomes to the TGN, either to escape lysosomal degradation or to gain access to the nucleus. One of the best examples is the interaction of human papillomavirus (HPV) with SNX17 [49]. Here the PDZ domain of SNX17 and SNX27 interacts with the viral capsid protein L2 and enhances HPV infection by trafficking L2 and the bound viral DNA from the late endosomes to the TGN and subsequently to the nucleus [223]. In a similar way, SNX2 traffics the human respiratory syncytial virus (HRSV) structural proteins to enhance viral infection [224]. Consistent with this, various studies have shown that deletion of specific retromer components inhibits specific steps in the intracellular life cycle of the vaccinia virus, hepatitis C virus (HCV), and human papillomavirus (HPV) [225]. Other pathogens have also evolved elegant mechanisms to inhibit the innate immune response roles of SNXs. SNX5 and SNX6 are inhibited by *Legionella* RidL and *Chlamydia* IncE to evade the lysosomal mediated degradation [226]. It would be of great interest to see if SARS-CoV-2 and other pathogenic viruses regulate SNX trafficking activity to enhance viral progeny production or evade the innate immune response.

## 9. Conclusions

Cells sense and respond to various internal and external stimuli to regulate processes such as gene expression, cell cycle progression, metabolism, and protein homeostasis. In the cell, there are severe quality control mechanisms held in place to regulate protein degradation. The mode of protein degradation depends on several factors including size, localization, and timing of substrate proteolysis. For example, the large size of organelles and multiple subunit complexes requires lysosomal degradation. Localization of proteins such as transmembrane proteins requires lysosomal degradation because these proteins are embedded within the membrane making proteasomal-mediated degradation unfavorable. For the cell to quickly turn genes on and off transcription factors are degraded via nuclear 26S proteasomes. This mode of degradation, therefore, relies on spatiotemporal factors because degradation needs to happen rapidly, and proteasomes are localized in close proximity within the nucleus. Understanding the molecular details behind SNX cargo recognition, membrane binding, and protein degradation will provide insight into the diverse roles of SNXs in various biological processes. The growing evidence of SNXs in the three quality control protein homeostasis pathways will shed light on pathologies associated with perturbed proteolysis and provide innovative targets for therapeutics.

## Figures and Tables

**Figure 1 cells-10-00017-f001:**
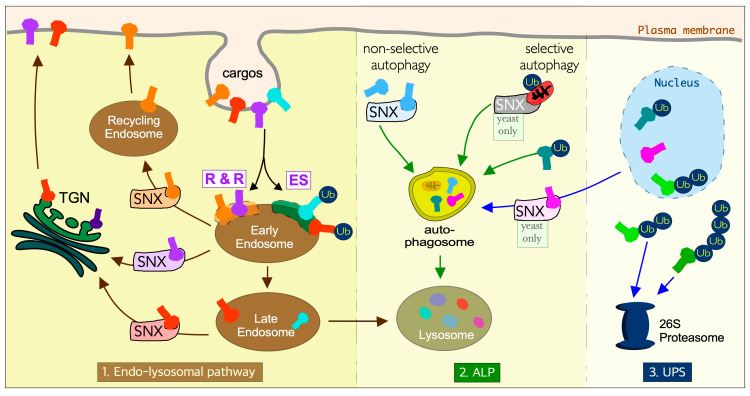
Sorting nexins and ubiquitin coordinate the three distinct but interconnected protein proteolysis pathways. In the endo-lysosomal pathway, membrane proteins are sorted at the early endosome. Cargos destined for lysosomal degradation are marked by ubiquitination and internalized using endosomal sorting complexes required for transport pathways (ES). Cargos destined for recycling are retrieved either by the retromer or retrieval complex (R & R) coupled with various subclasses of sorting nexins (SNX- see text for details). In the autophagy-lysosome pathway (ALP) cargos are sequestered to the vacuole by double-membraned vesicles called autophagosomes by selective or non-selective mechanisms. Selective pathways in yeast are mediated by SNX-Bar heterodimers. In mammalian cells, the recognition of selective autophagy cargos is dependent upon ubiquitination (Ub). The ubiquitin-proteasomal system (UPS) targets short-lived regulatory proteins that are selectively targeted and degraded. TGN- *trans*-Golgi network, SNX-sorting nexin. In cells, the TGN and nucleus are in close proximity, whereas here they are drawn apart for clarity.

**Figure 2 cells-10-00017-f002:**
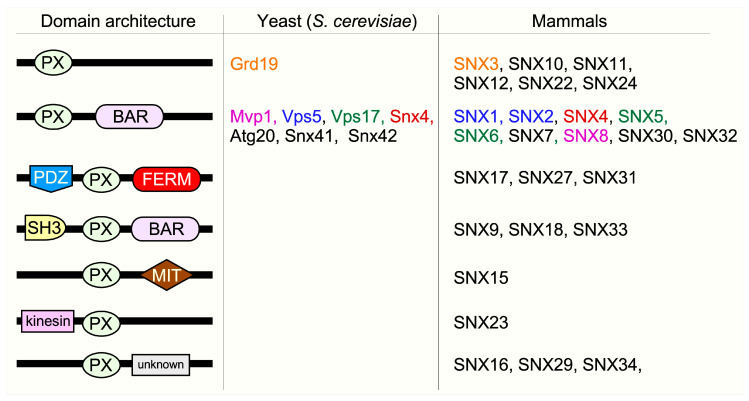
Structural classification of SNX subfamilies. The domain architecture describes the functional domains within different SNX subfamilies. The phox homology (PX) domain denotes the highly conserved lipid-binding domain that unifies the SNX protein family. It enables SNXs to bind to specific phosphoinositides (PtdIns) which mark different membrane surfaces. SNX-BAR (Bin/Amphiphysin/Rvs) proteins contain coiled-coil regions that enhance membrane binding, membrane remodeling, and protein-protein interactions. PDZ (postsynaptic density 95/discs large/zonula occludens) domains, FERM (protein 4.1/ezrin/radixin/moesin) domains, SH3 (SRC homology 3) domains, MIT (microtubule interacting and trafficking) domains, and Kinesin motor domains play a role in membrane binding, substrate recognition, kinase activity regulation, protein trafficking, and binding/movement along microtubules. The color code signifies yeast and mammalian homologs.

**Figure 3 cells-10-00017-f003:**
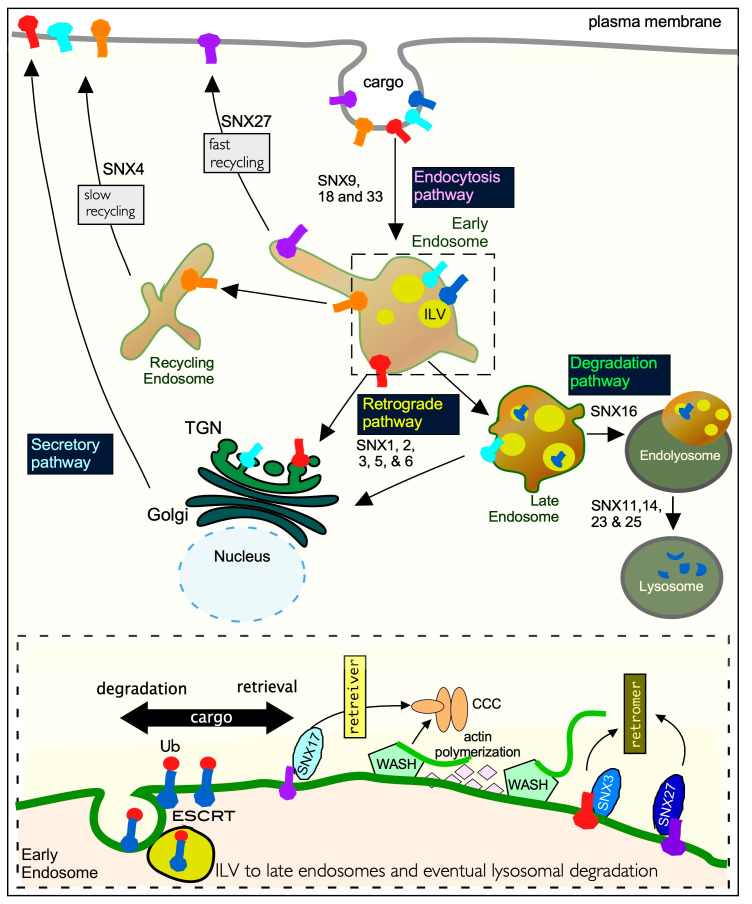
The role of SNXs in mammalian endocytosis. Transmembrane proteins are internalized into early endosomes from the plasma membrane by the endocytosis pathway. From here they are sorted through the complex and dynamic endomembrane network that consists of three different endosome vesicles (early, late, and recycling). Retrograde transport of membrane proteins requires recycling endosomes and the *trans*-Golgi network (TGN) for delivery back to the plasma membrane by the secretory pathway. Degradation of membrane proteins requires multiple rounds of cargo sorting and intra-luminal vesicle (ILV) biogenesis and fusion of the late endosome with the lysosome to form endo-lysosomes. Here ILVs and their accompanying cargos are degraded. The traced box at the bottom of the diagram is a zoomed-in schematic of sorting at endosomal membranes. It indicates that ubiquitinated membrane proteins destined for lysosomal proteolysis mediated by the ESCRT pathway. Transmembrane proteins are sorted by retromer and retrieval complexes aided by the actin remodeling WASH complex. Only SNX17 associates with the retriever and cooperates with the CCC complex to mediate endosomal trafficking. The retromer complex controls the recycling of a wide range of different cargos in cooperation with multiple SNX proteins including SNX3 and SNX27. Ub—Ubiquitin; CCC—CCDC22, CCDC93, and COMMD; WASH—Wiskott–Aldrich syndrome protein and SCAR homolog; ESCRT—endosomal sorting complexes required for transport.

**Figure 4 cells-10-00017-f004:**
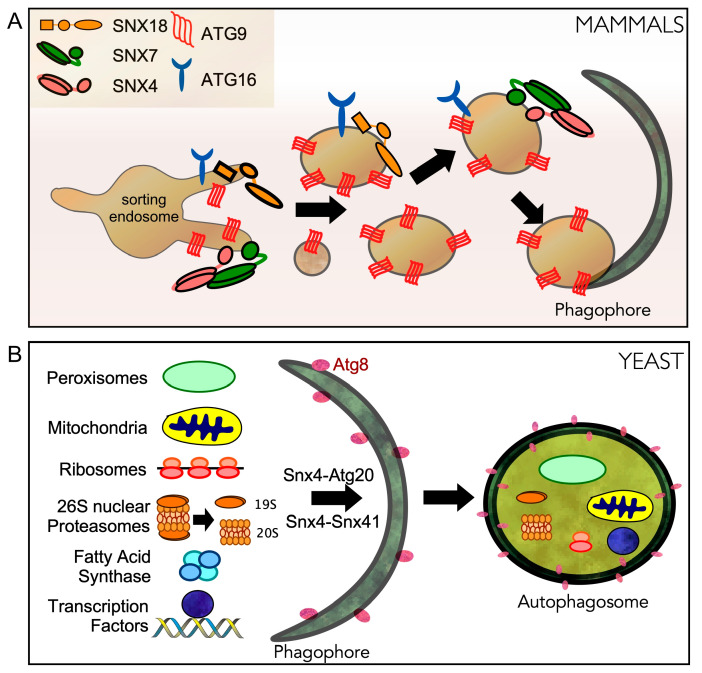
Sorting nexins play critical roles in autophagy following starvation stress. (**A**) In mammalian cells, SNX4/SNX7 and SNX18 are required for autophagy-dependent localization of ATG16 and ATG9 from recycling endosomes to the pre-autophagosomal site (PAS). ATG9 containing vesicles are required for PAS formation and autophagosome biogenesis. (**B**) In yeast, Snx4-Atg20 (Snx42) and Snx4-Snx41 heterodimers regulate different forms of selective autophagy.

**Figure 5 cells-10-00017-f005:**
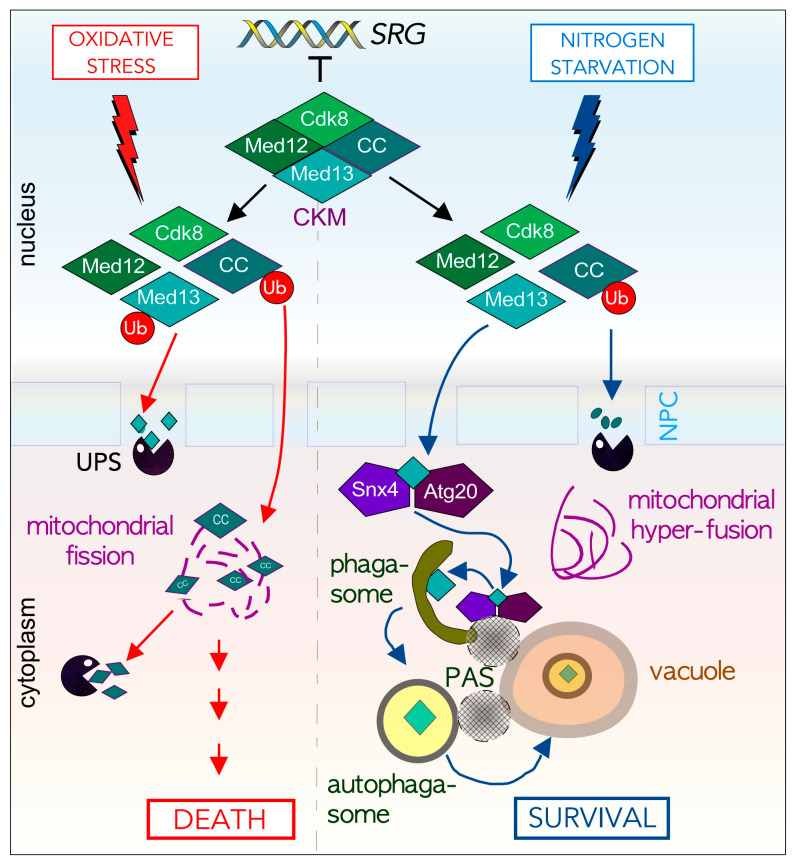
Med13 is destroyed either by the UPS or Snx4-Atg20 mediated autophagic degradation following cell death (left) or survival signals (right). Cartoon outlining stress-dependent fates of cyclin C and Med13, two members of the Cdk8 Kinase module (CKM). Here the subcellular address of cyclin C following stress mediates cell fate decisions by affecting mitochondria morphology. See text for details and [111,171,172,181].

**Table 1 cells-10-00017-t001:** List of mammalian sorting nexins and associated proteins (WASH) that play a role in the UPS.

SNX	UPS Activity	Role	Ref
SNX3	ubiquitin-specific protease 10 (USP10)	Deubiquitylates and stabilizes SNX3	[129]
SNX5	FBW7	Interacts with FBW7 and blocks FBW7-mediated ubiquitination of oncoproteins such as c-Myc, NOTCH1, and Cyclin E1	[130]
SNX9	Itch (atrophin-1 interacting protein 4, Nedd family member)	Itch regulates intracellular levels of SNX9	[126]
SNX16	indirect	Postulated that SNX16 interacts with and inhibits proteasome-dependent ubiquitination of eukaryotic translation elongation factor 1 A2 (eEF1A2), thereby activating c-myc signaling.	[131]
SNX18	MIBd1 E3 ligase	Promote the endocytosis of Delta-like protein 1 (Dll1) which is the transmembrane ligand protein for the Notch proteins.	[125]
SNX27	Non-catalytic role of the deubiquitinase OTULIN	OTULIN antagonizes SNX27-dependent cargo loading and binding to the VPS26A and affects endosome-to-plasma membrane trafficking.	[132]
retromer	TRIM27 E3 ubiquitin ligase (non-catalytic role)	Mediates the phosphorylation and activation of STAT	[133]
retromer	MAGE-L2-TRIM27 E3 ubiquitin ligase	The MAGE-L2-TRIM27 E3 ubiquitin ligase localizes to retromer-positive endosomes.	[134]
WASH	K63-linked ubiquitination and deubiquitinase USP7	WASH is activated by K63-linked ubiquitination of WASH K220 by MAGE-L2-TRIM27. USP7 regulates this activity.	[128,134]

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
