# Peer review of "Sorting Nexins in Protein Homeostasis"

_cells, 2020, doi:10.3390/cells10010017_

Round 1

Reviewer 1 Report

This review by Hanley and Cooper addresses the role of sorting nexins in maintaining protein homeostasis, highlighting their role in targeting proteins to different protein quality control systems. The authors have focused on an interesting topic that could be beneficial to the researchers who want to explore or work in this field. In my opinion the content of the manuscript is not particularly focused on the topic of the special issue which concerns the crosstalk between the ubiquitin and autophagy systems.  

The manuscript must be deeply edited for typos, incomplete sentences, repetition which make reading of the paper difficult, especially the last part concerning diseases.

I have some suggestions which may help improvise the quality of the manuscript

Abstract

I suggest revising the abstract to include the conclusions from that review rather than the topic of the review. Lines 16-17, how SNX take part to the other proteolytic systems?

Line 11, what is cellular sorting, did you mean protein sorting and trafficking? Sentence in lines 19-21 makes no sense.

Figure legends

The legend of Figure 1 describes the three main proteolytic pathways, but it does not emphasize the role of SNX within them. A more detail explanation of what it is represented is necessary. As a minimum the figures should have legends that define all abbreviations used.

The same observation applies to all other figure legends, there is not a clear explanation of what it is depicted in the figure. The legend should be informative

Text organization

I have problems to understand text organization. Please consider starting with SNX classification after the introduction and then move to the role of SNX in the endocytosis/lysosomal system, in the ubiquitin/proteasome and in the autophagy lysosomal pathways. Please consider changing the title of paragraph 2.2, it is the same of the review, should be more focused on the content of the paragraph

I believe that the title “Ubiquitin and Endocytosis” should be changed as the content of the following text concerns the interplay between the ubiquitin proteolytic system and SNXs.

The part concerning coronavirus is completely out of the focus since, as stated by the authors, there is no information available on the interplay between coronavirus and SNX; I think it should be deleted because too speculative in this context

Other observations

Lines 575-580, it is impossible to understand the meaning

Line 113, …to or by degradative pathways?

Line 122, facilitate the fate is too generic, please specify

Lines 167-168, please consider changing the sentence with: The SNX-PX subfamily consists only of a PX domain. This family includes SNX3 which forms with the core retromer (VPS26, VPS29, and VPS35) a multi-protein complex named the SNX3-retromer. This complex …….Lines 170-171 should be deleted because contain the same concept.

Lines 229-230 it is not clear

Lines 316, please correct

Lines 483-490, delete it is a repetition of the previous paragraph

Line 501, please rephrase the title, the SCFgrr1 is mentioned here but not discussed in the text, what is it? What is its involvement in the processes described below?

Lines 538-539 SNX6-mediated endolysosomal degradation?

Lines 540-541, to which proteolysis systems do the authors refer?

Lines 544-545, deficiencies, or dysregulation of sorting nexins results in protein homeostatic or in disruption of protein homeostasis?

Lines 563-564, check the sentence, the same for the sentence below ……..regulate insulin degradation

Lines 579-580, check

Author Response

Thank you very much for reviewing our manuscript. The revised manuscript has been significantly updated, guided by reviewer’s comments and advise. We very much appreciated your input and would sincerely like to apologize for the huge numbers of typos. There is no excuse, and we hope that you accept our sincere apologies.

We have taken your valid concerns into consideration and updated the manuscript accordingly. Specific answers to your questions are given below.

Firstly, we rewrote the abstract to include the conclusions from the review rather than the topic of the review. Secondly, we edited all the figure legends, ensuring that all the abbreviations were defined. The text organization was changed, and we started with SNX classification after the introduction and thereafter moved to the role of SNX in the endocytosis/lysosomal system. We changed the title to “Sorting nexins in protein homeostasis” as we feel that this better  reflects the contents of the review. Lastly, we removed the coronavirus section.

Again, we thank you for taking the time to review our manuscript and we hope that you now find it acceptable for publication.

Reviewer 2 Report

Thank you for the invitation to review this manuscript. I hope that our comment and suggestion will be helpful to improve the review, which I found excellent. Please find here minor comments.

In this review the authors summarize and discuss the implication of sorting nexins in proteostasis pointing their roles in the coordination of key processes such as protein sorting autophagy-lysosome mediated degradation and UPS/ubiquitin proteasome system. The review is very well written, concise and supporting figure are well designed. The author succeeds to discuss an impressive number of high-quality references related to the field. I really congratulate them for this high-quality review.

I have some minor “suggestions about the structure:

1- Is it possible for the authors to start by part 3 (Classification of SNX proteins) and then paragraph 2 (Outline of endocytosis pathways)? I think that this could be helpful for the large audience to have an overview on the SNXs family before their cellular functions.

2- In part 4 (Snx cargo recognition in yeast), as indicated in the title the authors concentrate on Yeast. Is it because there are no or very few existing data in higher eucaryotes? Actually, in this paragraph the authors expose and discuss many works performed in other systems. In my point of view part 4 should be entitled (Retromer dependent and independent sorting nexin function) where the author could comment in parallel how SNXs impacts this machinery in many systems. The first part 4.1 could be integrated in ex part 2 (Outline of endocytosis pathways -SNXs in endocytic system).

3- Part 5 (an excellent part of the review) contains many sub-titles with sometime short and brief information. I propose to reformat this part in a single paragraph that could be entitled “Sorting Nexins regulation of UPS arms in endocytic system”. Also, paragraph 5.5, which discuss oncogenic role of SNXs-UPS regulation, could be transferred to part 8 (Sorting nexins in disease).

4- Part 6 is also a very interesting part of the review could be reformatted as follow:

I think a two sentences introduction of autophagy mechanisms is sufficient in paragraph 6.2. There is no need to have an independent part (6.1). Then the two paragraphs 5.5 and 6.6 seems to be out of the review scoop as the authors did not comment any evidences about a direct implication of SNXs in mTORC1 signaling. I propose to delete these two paragraphs.

Part 7 and 8 are excellent and well structured.   

Author Response

Thank you very much for reviewing our manuscript. The revised manuscript has been significantly updated, guided by reviewer’s comments and advise. We very much appreciated your input and would sincerely like to apologize for the huge numbers of typos. There is no excuse, and we hope that you accept our sincere apologies.

We have taken your valid concerns into consideration and updated the manuscript accordingly. Specific answers to your questions are given below.

We have taken your valid concerns into consideration and revised the updated manuscript accordingly. Firstly, at your suggestion the text organization was changed, and we started starting with SNX classification after the introduction and thereafter moved to the role of SNX in the endocytosis/lysosomal system. In section 2 and 3 in the revised manuscript we outline in detail the retromer dependent and independent function in mammalian cells. Section 4 has been heavily edited to mainly contain information about yeast. At your suggestion we moved section 5.5 on oncogenes to the section on diseases. Thank you for this suggestion as this is a much better placement. At your suggestion we also consolidated the rest of section 5. We made all the edits that you suggested in section 6 as well as adding in some new and exciting data from Beth Levine group on SNX5 dependent virus-induced autophagy that is independent of TORC1.

Again, we thank you for taking the time to review our manuscript and we hope that you now find it acceptable for publication.

Reviewer 3 Report

Cooper and coworker summarized the role of sorting nexins in protein homeostasis, both in yeast and mammals. This is quite a comprehensive review that covers the connection among SXNs, UPS, autophagy, and endocytosis. Sections are organized clearly to the point. It is highly recommended for acceptance after addressing a few minor questions. 

  1. In Section 2, during endocytosis, either the recycling or the degradation pathways, is there any research evidence that have shown certain cargos remain in the cytosol during these processes? 
  2. Line 618, could the authors further explain the process of CoVs releasing their RNA contents, e.g. what is the mechanism of RNA release for these viruses? Where does the RNA occur? Does it go through an endosomal escape process?

Author Response

Thank you very much for reviewing our manuscript. The revised manuscript has been significantly updated, guided by reviewer’s comments and advise. We very much appreciated your input and would sincerely like to apologize for the huge numbers of typos. There is no excuse, and we hope that you accept our sincere apologies. Specific answers to your questions are given below.

  1. In endocytosis, cargos are processed between endosomes and the TGN or taken to lysosomes via ILVs for degradation. Thus, their exposure to the cytosol is limited and I did not come across literature that discusses this. However, in ubiquitin mediated degradation and ALP however, cargoes are often cytosolic. I hope that this answers your question.
  2. In the revised manuscript, at the suggestion of other reviewers, we have taken out the section on CoV’s. However, other it is known that other viruses do use fast recycling pathways – this is talked about in the review.

Again, we thank you for taking the time to review our manuscript and we hope that you now find it acceptable for publication.

Round 2

Reviewer 1 Report

I have found the revised version of the manuscript by Hanley & Cooper significantly improved compared to its initial version. My concerns have mostly been addressed and the text modified accordingly.

I have carefully read the text again and I have only few minor suggestions/questions before publication. 

Lines 25-27. Since only short-lived proteins have regulatory roles, I suggest rephrasing as follows: “the UPS is the primary proteolytic route for misfolded or damaged proteins, and short‐lived proteins having essential functions in many critical cellular pathways, including cell cycle progression and transcriptional regulation”

Lines 53-54. “Although initially reported as independent pathways, these systems are now known to be interconnected, linked by their common requirement for ubiquitin in substrate targeting.” Please specify to which systems you refer instead of saying “these system”

Figure 1. According to the title, this figure should highlight the role of Ub and SNX in the three main proteostasis pathways. While it is clear the role of Ub, less clear is the involvement of SNXs.  SNXs are depicted in the panels; thus, their role within each pathway should be defined in the legend. Moreover, do SNXs recognize ubiquitinated cargo as depicted in the picture? (light blue cargo). I still do not understand why in the third panel ubiquitinated proteins and SNX-transported proteins exit only from the nucleus. Are these substrates only nuclear? Moreover, what is the involvement of SNX in this pathway; please indicate at least some of the interconnections between the Ub pathway and SNXs as discussed later in the text.

Lines 110-11. I am not sure which is a member of the SNX-PX family, the SNX3-retromer or SNX3 (a component of the retromer complex)?

Line 134. Please cite in the text Figure 3, where SNX17 and SNX27 are shown together with the retromer and retriever complexes.

Line 220, check apetaly

Figure 3. I am confused by the legend “The traced box at the bottom of the diagram is a zoomed‐in schematic of the different retrieval complexes (retromer, retriever, CCC, and WASH) used at the plasma membrane to engulf cell surface membrane proteins”, but in the text the authors state that the retriever and retromer complexes are associated to the endosomal membrane as it should be if their role is to rescue proteins from lysosome degradation (see picture in Current Biology 2017; 27:R1233)

Line 235, only in the recycling or recycling and degradative pathways?

Line 375. I would add a sentence highlighting that the UPS in turn regulates the levels/activity of SNXs. Since the regulation is reciprocal, I suggest changing the title of par. 5.2 accordingly.

Author Response

Thank you for taking the time to read this review again. We really appreciate your input. All your suggestions have been incorporated into the second revised draft.